# *Cerogamasus*, a New Genus of Parasitinae Mites, with Description of Four New Species from China (Acari: Parasitiformes: Parasitidae)

**DOI:** 10.3390/ani14152260

**Published:** 2024-08-03

**Authors:** Maoyuan Yao, Jianxin Chen, Tianci Yi, Daochao Jin

**Affiliations:** 1College of Agriculture, Anshun University, Anshun 561000, China; maoyuanyao723@163.com (M.Y.); jianxinchen000@163.com (J.C.); 2Institute of Entomology, Guizhou University, Guiyang 550025, China; 3Guizhou Provincial Key Laboratory for Plant Pest Management of Mountainous Region, Guiyang 550025, China

**Keywords:** classification, new genus, new species, South China

## Abstract

**Simple Summary:**

The present study describes a new genus, *Cerogamasus* gen. nov., and four new species, *Cerogamasus tibetensis* Jin & Yao sp. nov., *Cerogamasus anhuiensis* Jin & Yao sp. nov., *Cerogamasus guizhouensis* Jin & Yao sp. nov. and *Cerogamasus multidentatus* Jin & Yao sp. nov. *Cycetogamasus coreanus* Athias-Henriot, 1980, is transferred from the genus *Cycetogamasus* to *Cerogamasus* gen. nov. An identification key to the known species of new genus is provided.

**Abstract:**

The new genus, *Cerogamasus* gen. nov., with the type species *Cerogamasus tibetensis* sp. nov., is established. The new genus is easily distinguished from other genera of Parasitidae because the dorsal idiosoma in both sexes bears more than 40 pairs of setae, of which fewer than 7 pairs of podonotal setae are smooth; the seta *z5* of the dorsal hexagon is similar to *j5* and *j6* in form (pilose or distally pilose) while different in length (*z5* longer); the seta *al* of the palpfemur is pectinate, and *al1* and *al2* of the palpgenu are entire; the gnathotectum is trispinate; peritrematal shields in females are posteriorly free; and the palptrochanter in males has a pointed ventral protuberance. *C. anhuiensis* sp. nov., *C. guizhouensis* sp. nov. and *C. multidentatus* sp. nov. are described based on adult samples; *C. tibetensis* sp. nov. is described based on deutonymph and adult samples. *Cycetogamasus coreanus* Athias-Henriot, 1980, is transferred to *Cerogamasus* gen. nov. as a new combination.

## 1. Introduction

The Mesostigmata is a large mite order corresponding to about 20% of all known mite species [1]. The family Parasitidae Oudemans, 1901, is among the most common and widely distributed families of Mesostigmata [1,2]. The mites of the family occur in soil and decaying organic material such as manure, debris and compost; some species are found on the body surface of birds or arthropods and in carcasses of mammals [3,4,5]. They are essentially predatory and feed upon other microarthropods and nematodes, including their eggs [3,6]. Parasitidae comprises more than 500 species and 46 genera in two subfamilies: Parasitinae Oudemans, 1901 (23 genera), and Pergamasinae Juvara-Bals, 1972 (23 genera) [7,8,9,10].

In a study of Parasitidae by Chinese scholars first reported in 1963, Jiang et al. [11] discovered *Poecilochirus necrophori* Vitzthum, 1930, and *P*. *subterraneus* Miiller, 1860, on the body surface of mice. In the following 30 years, the study of Parasitidae in China was slow; 22 species were reported [12,13,14,15]. Since the 1990s, many scholars have carried out surveys of Parasitidae in the Chinese fauna, and a large number of new species and new record species of Parasitidae have been discovered and reported [16,17,18,19,20,21,22,23,24,25,26,27]. The known Chinese fauna of Parasitidae has now grown to more than 150 species, nearly two-thirds of which have been reported only in China [27].

Four new species were found during a current study of the Chinese Parasitidae. According to the key for the families of the order Mesostigmata [22] as well as the definitions provided by Evans and Till [14] and Hrúzová and Fenďa [5], we have placed them in the subfamily Parasitinae. The aim of this study is to describe a new genus, *Cerogamasus*, and four new species, *C. tibetensis*, *C. anhuiensis*, *C. guizhouensis* and *C. multidentatus*, of Parasitidae from China and thus contribute to the knowledge of the fauna of Mesostigmata in Asia.

## 2. Materials and Methods

Individuals of four new species were extracted from decaying leaves, moss, weed piles or rotten wood from Berlese–Tullgren funnels for 12–24 h; placed in 75% alcohol; cleaned in Nesbitt’s solution; and then mounted on slides in Hoyer’s medium (distilled water/arabic gum/chloral hydrate = 5:3:20:2) for later identification [1]. All specimens were deposited in the Institute of Entomology, Guizhou University, Guiyang, China (GUGC).

Line drawings were prepared with the aid of a drawing tube attached to a phase-contrast Nikon Ni E microscope with DIC optics; photographs were taken using a camera (Nikon DS-Ri 2) attached to a Nikon Ni E microscope with DIC optics and figures were edited with Adobe Photoshop CC 2021. Measurements were carried out on all available specimens, the measuring followed that in [25] and all measurements were given in micrometres (μm).

The system of idiosomal setal nomenclature followed [3]. Terminology for the palp chaetotaxy followed [28], and then adenotaxy (idiosomal glands) and poroidotaxy (lyrifissures) followed [29,30]. The description of the males and deutonymphs omitted the features common with the females.

## 3. Results

Family Parasitidae Oudemans, 1901;Subfamily Parasitinae Oudemans, 1901; Genus *Cerogamasus* Jin & Yao gen. nov.; Type species *Cerogamasus tibetensis* Jin & Yao sp. nov.

The diagnosis of *Cerogamasus* gen. nov. in both sexes: dorsal idiosoma well sclerotized and reticulated, bearing more than 40 pairs of setae, of which fewer than seven pairs of podonotal setae are smooth; seta *z5* of the dorsal hexagon similar to *j5* and *j6* in form (pilose or distally pilose), but different in length (*z5* longer); seta *al* of palpfemur pectinate and *al1* and *al2* of palpgenu entire; corniculi short; gnathotectum three prongs (but four prongs in the male of *C. multidentatus* sp. nov.); fixed digits of chelicera with more than seven teeth.

In the female: podonotal and opisthonotal shields separate; metasternal shield separated from sternal and epigynal shields; epigynal shield subtriangular and separated from opisthogastric shield; opisthogastric shield bearing at most ten pairs of ventral setae; peritrematal shields posteriorly free; movable digits of chelicerae with more than seven teeth.

In the male: Holodorsal shield with a transverse suture in central region; the base of tritosternum reduced; the venter of hypostome elevated forward and forming a protuberance; setae *h1*, *h2* and *h3* on the protuberance, seta *pcx* near the base; palptrochanter with a pointed and horn-like ventral protuberance. Femur II with a main spur (proximal) and an axillary process (distally), genu II with or without spur, and tibia II with a spur.

**Etymology.** The name of the genus is derived from “cero–“, meaning “horn”, and refers to the horn-like process on the palp trochanter in males, with “–gamasus” referring to gamasine mites (masculine).

**Differential diagnosis:** The taxonomy of Parasitidae is based on some key morphological charactistics to separate genera. These include peritrematal shields of female posteriorly free or not; seta *al* on palpfemur entire or divided apically; both setae *al1* and *al2* on palpgenu entire or only one seta entire; metasternal shields of female detached from sternal and epigynal shield or fused; epigynal shield separated from opisthogastric shield or fused; male palptrochanter with or without ventral protuberance, venter of hypostome elevated forward and forming protuberance, or without; and genu II with or without a spur [3,7,8,9,10,11]. Considering these characteristics, *Cerogamasus* gen. nov. is similar to *Trachygamasus* Berlese, 1904; *Psilogamasus* Athias-Henriot, 1969; and *Coprocarpais* Hrúzová & Fenďa, 2018 [7,10,25,26].

*Cerogamasus* gen. nov. shares with *Trachygamasus* [25,26] the following characteristics: setae *al1* and *al2* on palpgenu entire, gnathotectum trifid, seta *z5* pilose or distally pilose, male genu II with or without spur and venter of hypostome elevated forward and forming a protuberance. However, *Cerogamasus* gen. nov. is different from *Trachygamasus* as follows: (1) seta *al* on palpfemur pectinate, vs. entire in *Trachygamasus*; (2) dorsal setae *j5* and *j6* pilose or distally pilose, vs. smooth in *Trachygamasus*; (3) female peritrematal shields posteriorly free, vs. fused to ventral shield in *Trachygamasus*; (5) female opisthonotal shield with more than 15 pairs of pilose setae, vs. fewer than 5 pairs in *Trachygamasus*; (6) male palptrochanter with a pointed ventral protuberance, vs. without protuberance in *Trachygamasus*.

*Cerogamasus* gen. nov. shares with *Psilogamasus* [10,31] the following characteristics: setae *al1* and *al2* of palpgenu entire, seta *al* of palpfemur pectinate, female peritrematal shields free posteriorly, gnathotectum trifid. However, the new genus is different from *Psilogamasus* as follows: (1) setae *z5*, *j5* and *j6* pilose or distally pilose, vs. smooth in *Psilogamasus*; (2) presternal platelets well defined, vs. absent in *Psilogamasus*; (3) seta *ZV1* present, vs. absent in *Psilogamasus*; (4) three prongs of gnathotectum pointed distally, vs. central prong apically with two to five branches or multidentate, lateral prongs spiculate with bifid apex or two to four apical branches in *Psilogamasus*; (5) podonotal shield of female with 22 pairs of setae, vs. 16–18 pairs in *Psilogamasus*; (6) opisthonotal shield with more than 20 pairs of setae, vs. 5–6 pairs in *Psilogamasus*; (7) male palptrochanter with one pointed ventral protuberance, vs. without protuberance in *Psilogamasus*.

*Cerogamasus* gen. nov. shares with *Coprocarpais* [7] the following characteristics: setae *al1* and *al2* on palpgenu entire, gnathotectum trifid, seta *z5* pilose or distally pilose, male palptrochanter with a big ventral protuberance. However, *Cerogamasus* gen. nov. is different from *Coprocarpais* as follows: (1) seta *al* on palpfemur pectinate, vs. entire in *Coprocarpais*; (2) dorsal setae *z5*, *j5* and *j6* pilose or distally pilose, vs. smooth in *Coprocarpais*; (3) female peritrematal shields posteriorly free, vs. fused to ventral shield in *Coprocarpais*; (4) podonotal shield of female with more than 12 pairs of pilose setae, vs. fewer than 5 pairs in *Coprocarpais*; (5) dorsal shields with flat reticulation, vs. distinct foveate sculpture in *Coprocarpais*; (6) male *al1* and *al2* acicular (button-shaped in *C. multidentatus* sp. nov.), vs. flat blades in *Coprocarpais*.

### 3.1. New Species

#### 3.1.1. *Cerogamasus tibetensis* Jin & Yao sp. nov.

**Diagnosis.** Both sexes: dorsal setae *z1*, *s1*, *s2*, *s6*, *r4*, *r5* and *r6* smooth; gnathotectum emerging from nude base; seta *pcx* on gnathosoma smooth; setae *v2* on palptrochanter stouter than *v1*. In the female, endogynium with an inverted V-shaped structure, each side of which is flanked with a reniform structure, and the rear with a saccate structure; opisthogastric shield bearing ten pairs of ventral setae, of which one pair thick and distally pilose. In the male, opisthogastric region with two pairs of distally pilose setae; movable digit with a blunt tooth; genu II and tibia II with a spur.

**Type material.** Holotype: ♀, encountered in moss on the wood, Bomi County (2624 m a.s.l., N 29°54′34″, E 95°29′37″), Linzhi, Tibet Autonomous Region, China, 17 July 2019. Paratypes: 1 ♀ and 1 deutonymph with the same details as holotype; 2 ♀♀ and 2 ♂♂ encountered in decaying leaves, Bomi County (2535 m a.s.l., N 29°56′44″, E 95°23′12″), Linzhi, Tibet Autonomous Region, China, 18 July 2019.


**Description**


**Female** (n = 4). *Dorsal idiosoma* (Figure 1A). Idiosoma well sclerotized; length: 774–793; width: 561–588. Podonotal shield with 22 pairs of setae, of which setae *z1*, *s1*, *s2*, *s6*, *r4*, *r5* are fine and smooth, others are thick and pilose. Lengths of setae on podonotal shield: *j1* 78–81, *j2* 94–96, *j3* 94–98, *j4* 97–100, *j5* 72–75, *j6* 65–68, *z1* 48–51, *z2* 65–68, *z3* 77–81, *z4* 82–86, *z5* 89–92, *z6* 54–56, *s1* 20–23, *s2* 14–15, *s3* 85–87, *s4* 86–89, *s5* 86–88, *s6* 35–37, *r2* 70–73, *r3* 161–167, *r4* 17–19, *r5* 24–27. Seta *r6* is smooth and off the shield. Opisthonotal shield with 22 to 24 pairs of thicked and pilose setae, of which the setae *R* series is shorter than the *J*, *Z* and *S* series.

*Ventral idiosoma* (Figure 1B). Tritosternum flanked by three pairs of presternal platelets close to each other. Sternal shield bearing three pairs of setae (*st1*–*st3*) and two pairs of poroids (*iv1*–*iv2*). Metasternal shield, bearing seta *st4* and *iv3*, separated from sternal and epigynal shields. Gland pores *gv2* not seen. Endogynium (Figure 1C) with an inverted V-shaped structure, its sides flanked with a reniform structure, and the rear with a saccate structure. Opisthogastric shield bearing ten pairs of ventral setae, seta *JV4* thick and pilose. Opisthogastric soft cuticle with five pairs of setae, *JV5* (70–72) pilose and longer than others. Peritrematal groove length 336–381. Lengths of setae on ventral shields: *st1* 72–75, *st2* 70–73, *st3* 65–68, *st4* 54–57, *st5* 54–55, *JV1* 44–46, *JV2* 57–59, *JV3* 55–57, *JV4* 63–65, *ZV1* 17–20, *ZV2* 43–45, *ZV3* 46–47, *ZV4* 41–43, *SV1* 19–21, *SV2* 36–38, *pa* 27–29, *po* 27–30.

*Gnathosoma* (Figure 1D–G). Gnathotectum with three prongs. Fixed and movable digits of chelicera with several small teeth. Deutosternal groove with 12 denticulate rows; setae *h1* (79–81), *h2* (73–75), *h3* (68–70) and *pcx* (82–85) smooth. Palp length: 274–305; seta *v2* on palptrochanter stouter than *v1*; seta *al* on palpfemur pectinate; *d1* and *d2* distally pilose; *al1* and *al2* on palpfemur spatulate, others simple.

*Legs.* Lengths of legs: I 796–919, II 649–666, III 553–612, IV 939–992. Leg II stouter than other legs. Setae *av1*, *al1*, *pv1* and *pl1* on tarsi II–IV modified to short and thick spurs. Most leg setae distally pilose (Figure 2).

**Male** (n = 2). *Dorsal idiosoma* (Figure 3A). Idiosoma length: 715–757; width: 436–470. Holodorsal shield covering entire dorsum; a suture closely anterior to seta *J1*. All setae on the shield; setal form as in female.

*Ventral idiosoma* (Figure 3B). Tritosternum base reduced. Genital opening flanked by one pair of presternal platelets, sometimes fragmented. Lengths of setae *st1*–*st5* on sternogenital region: *st1* 67–70, *st2* 60–65, *st3* 52–56, *st4* 50–53, *st5* 50–52. Opisthogastric region with 14 pairs of setae, of which 6 pairs are distally pilose, *JV4* (92–94) and *JV5* (90–92), stouter and longer. Peritrematal groove length: 352–373.

*Gnathosoma* (Figure 3C–F). Palp length: 255–291; trochanter bearing seta *v2* evidently stouter and longer than *v1*. Movable digit with a blunt tooth. Lengths of setae *h1–h3* and *pcx* on hypostome: *h1* 55–58, *h2* 65–68, *h3* 72–75 and *pcx* 67–71.

*Legs.* Lengths of legs: I 824–893, II 612–676, III 544–591, IV 872–938. Femur II with a main spur (proximal) and an axillary process (distally); genu II and tibia II with a spur (Figure 3G).

**Deutonymph** (n = 1). *Dorsal idiosoma* (Figure 4A). Idiosoma weakly sclerotized; length: 710; width: 480. Podonotal and opisthonotal shields with 20 and 14 pairs of setae, respectively. Membranous cuticle of dorsal shields bearing 14 pairs of setae. Setae *z1*–*z4*, *z6*, *s1*–*s3*, *s6*, *r2*, *r4*–*r6* fine and smooth, others distally pilose. Seta *J1* absent. Lengths of setae on dorsal shields: *j1* 67, *j2* 68, *j3* 79, *j4* 91, *j5* 64, *j6* 59, *z1* 32, *z2* 55, *z3* 53, *z4* 77, *z5* 85, *z6* 39, *s1* 20, *s3* 60, *s4* 71, *s5* 75, *s6* 23, *r2* 48, *r3* 157, *r5* 19, *J2* 45, *J3* 41, *J4* 30, *J5* 44, *Z1* 72, *Z2* 48; *Z3* 41, *J4* 34, *S1* 31, *S2* 37, *S3* 31, *S4* 34.

*Ventral idiosoma* (Figure 4B). Tritosternum flanked by a pair of irregular presternal platelets. Sternal shield reticulated with a pale anterior transverse strip, bearing four pairs of setae (*st1*–*st4*), of which setae *st1* (55) and *st2* (51) are longer than *st3* (37) and *st4* (34). Opisthogastric region with 15 pairs of setae and a pair of metapodal shields. Anal shield reticulated. Setae *pa* and *po* equal in length (29–30). Peritreme groove length: 291.

*Gnathosoma* (Figure 4C–F). Gnathotectum emerging from denticulate base. Deutosternal groove with 11 denticulate rows. Lengths of setae: *h1* 64, *h2* 52, *h3* 62 and *pcx* 64. Palp length: 267; seta *v2* and *v1* on trochanter equal in size.

*Legs* (Figure 5). Lengths of legs: I 773, II 553, III 524, IV 821. Setae *av1*, *al1*, *pv1* and *pl1* on tarsi II–IV fine and setiform.

**Other stages**. Unknown.

**Etymology.** The new species name is derived from the type locality Tibet Autonomous Region, China (*tibetensis*). 

**Differential diagnosis.** The female of *C. tibetensis* sp. nov. is morphologically similar to *C. coreanus* comb. nov. in the setal form on the subcapitulum and opisthonotal shields, the number of teeth on the cheliceral fixed digit and the endogynium reniform structure [32,33]. However, it differs from *C. coreanus* comb. nov. as follows: (1) opisthogastric shield bearing ten pairs of setae, vs. five pairs in *C. coreanus* comb. nov.; (2) seta *v2* on palptrochanter smooth and stouter than *v1*, vs. distally pilose with setae *v1* and *v2* equal in size in *C. coreanus* comb. nov.; (3) deutosternal groove with 12 denticulate rows, vs. 8 rows in *C. coreanus* comb. nov.; (4) setae *SV1* present, vs. absent in *C. coreanus* comb. nov.

#### 3.1.2. *Cerogamasus anhuiensis* Jin & Yao sp. nov.

**Type material.** Holotype: ♀, encountered in decomposing leaves, Mount Huangshan Scenic Area (1785 m a.s.l., N 30°8′8″, E 118°9′38″), Huangshan, Anhui Province, China, 23 May 2018. Paratypes: four ♀♀ and three ♂♂ with the same details as holotype.

**Diagnosis.** Both sexes: dorsal setae *z1*, *s1*, *s2*, *s6*, *r4*, *r5* and *r6* smooth; gnathotectum emerging from denticulate base; seta *pcx* on gnathosoma pilose; setae *v1* and *v2* on palptrochanter equal in size. In females, endogynium with a saccate structure, its centre having two elliptic structures; opisthogastric shield bearing nine pairs of ventral setae, of which one pair of setae is distally pilose. In males, opisthogastric region with one distally pilose; movable digit with a blunt tooth; genu II without spur, tibia II with a spur.


**Description**


**Female** (n = 5). *Dorsal idiosoma* (Figure 6A). Idiosoma well sclerotized; length: 742–756; width: 504–551. Podonotal and opisthonotal shields with reticulation. Podonotal shield bearing 22 pairs of setae, of which setae *z1*, *s1*, *s2*, *s6*, *r4*, *r5* are fine and smooth, and others thick and pilose. Lengths of setae on podonotal shield: *j1* 72–75, *j2* 83–85, *j3* 91–95, *j4* 101–104, *j5* 63–65, *j6* 69–72, *z1* 52–55, *z2* 65–68, *z3* 84–86, *z4* 84–87, *z5* 83–85, *z6* 72–75, *s1* 28–31, *s2* 29–30, *s3* 81–83, *s4* 86–88, *s5* 88–90, *s6* 17–19, *r2* 81–84, *r3* 167–169, *r4* 19–21, *r5* 18–21. Seta *r6* (18–21) smooth and off the shield. Opisthonotal shield bearing 20 pairs of thick and pilose setae, of which setae *R* serie (50–55) shorter than *J* (60–70), *Z* (61–85) and *S* (58–83) series.

*Ventral idiosoma* (Figure 6B). Tritosternum flanked by two pairs of presternal platelets. Sternal setae *st1* (65–67) longer than *st2* (48–51) and *st3* (52–54). Seta *st4* length 44–46. Endogynium with a saccate structure, its centre having two elliptic structures (Figure 6C). Opisthogastric shield bearing nine pairs of ventral setae; seta *JV4* thick and pilose. Opisthogastric soft cuticle with five pairs of setae; *JV5* (48–50) thick and distally pilose. Gland pores *gv2* with three openings. Peritrematal groove length 302–316. Lengths of setae on opisthogastric shield: *JV1* 35–38, *JV2* 40–42, *JV3* 41–43, *JV4* 41–44, *ZV1* 12–14, *ZV2* 35–37, *ZV3* 46–49, *SV1* 13–16, *SV2* 21–24, *pa* 17–19, *po* 18–20.

*Gnathosoma* (Figure 6D–G). Gnathotectum with three prongs, emerging from denticulate base. Fixed and movable digits of chelicera with several small teeth. Deutosternal groove with 11 denticulate rows; setae *h1* (58–64), *h2* (46–51), *h3* (54–57) smooth, *pcx* (63–65) slightly pilose. Palp length: 265–281; trochanter bearing two stout setae (*v1* and *v2*). Femur with five setae of which *al* is pectinate, and others slightly pilose; genu with six pairs of setae, of which *al1* and *al2* are entire.

*Legs*. Lengths of legs I–IV: 844–877, 551–623, 512–581, 790–928, respectively. Most leg setae distally pilose.

**Male** (n = 3). *Dorsal idiosoma* (Figure 7A). Idiosoma length: 712–746; width: 492–546. Holodorsal shield covering entire dorsum, a suture closely anterior to seta *J1*. All setae on shield. Setal form as in female.

*Ventral idiosoma* (Figure 7B). Tritosternum base reduced. Genital opening flanked by two pairs of presternal platelets. Lengths of setae *st1*–*st5* on sternogenital region: *st1* 67–69, *st2* 49–54, *st3* 51–54, *st4* 45–48, *st5* 47–50. Ventrianal region with 14 pairs of setae, of which *JV5* (48–52) stout and pilose distally. Peritrematal groove length: 207–321.

*Gnathosoma* (Figure 7D–F). Gnathotectum emerging from nude base. Movable digit with a blunt tooth. Palp length: 265–281; trochanter with one pointed ventral protuberance, bearing short and fine setae *v1*, and *v2* and *v2* stouter than *v1*. Lengths of setae: *h1* 58–64, *h2* 46–51, *h3* 54–56 and *pcx* 63–65.

*Legs.* Lengths of legs I–IV: 844–877, 551–623, 512–623, 790–928, respectively. Leg II obviously stouter than others. Femur II with a main spur (proximal) and an axillary process (distally); tibia II with a spur (Figure 7G).

**Other stages.** Unknown.

**Etymology.** The new species name is derived from the type locality Anhui Province, China (*anhuiensis*). 

**Differential diagnosis.** *C. anhuiensis* sp. nov. is morphologically similar to *C. tibetensis* sp. nov. in the setal form with regard to dorsal shields, the number of teeth on the cheliceral and tibia II with a spur in males. However, *C. anhuiensis* sp. nov. female is different from *C. tibetensis* sp. nov. as follows: (1) opisthogastric shield bearing nine pairs of setae, vs. ten pairs in *C. tibetensis* sp. nov.; (2) gnathotectum emerging from denticulate base, vs. nude base in *C. tibetensis* sp. nov.; (3) seta *v2* on palptrochanter stouter than *v1*, vs. seate *v1* and *v2* equal in size in *C. tibetensis* sp. nov.; (4) seta *pcx* on gnathosoma pilose, vs. smooth in *C. tibetensis* sp. nov. The differences between them in the male are as follows: (1) opisthogastric region with one pair of setae distally pilose, vs. six pairs in *C. tibetensis* sp. nov.; (2) genu II without spur, vs. a spur in *C. tibetensis* sp. nov.

#### 3.1.3. *Cerogamasus guizhouensis* Jin & Yao sp. nov.

**Type material.** Holotype: ♀, encountered in weed pile, Tuanlong Village (1031 m a.s.l., N 27°54′52″, E 108°38′20″), Tongren, Guizhou Province, China, 1 May 2019. Paratypes: three ♀♀, same collection data as the holotype; two ♀♀, encountered in decaying leaves, Tuanlong Village (1304 m a.s.l., N 27°55′1″, E 108°38′21″), Tongren, Guizhou Province, China, May 1, 2019; one ♀ and three ♂♂ encountered in a weed pile, Tuanlong Village (1045 m a.s.l., N 27°54′5″, E 108°36′3″), Tongren, Guizhou Province, China, 2 May 2019.

**Diagnosis.** Both sexes: dorsal setae *z1*, *z2*, *s1*, *s2*, *r4*, *r5* and *r6* smooth; gnathotectum emerging from denticulate base; seta *pcx* on gnathosoma pilose; setae *v1* and *v2* on palptrochanter equal in size. In the female, endogynium with a saccate structure, its centre having a floriform structure; opisthogastric shield bearing ten pairs of ventral setae, of which four pair distally pilose. In the male, opisthogastric region with eight pairs of setae distally pilose; movable digit with a big blunt tooth and several small teeth; genu II and tibia II with a spur.


**Description**


**Female** (n = 7). *Dorsal idiosoma* (Figure 8A). Idiosoma well sclerotized, length 735–768, width 494–548. Podonotal shield with 22 pairs of setae, of which setae *z1*, *z2*, *s1*, *s2*, *r4*, *r5* fine and smooth, other thicked and pilose. Lengths of setae on podonotal shield: *j1* 62–63, *j2* 69–71, *j3* 67–70, *j4* 74–77, *j5* 52–55, *j6* 44–48, *z1* 37–40, *z2* 46–48, *z3* 59–61, *z4* 62–63, *z5* 65–67, *z6* 45–46, *s1* 27–29, *s2* 26–30, *s3* 56–58, *s4* 60–61, *s5* 60–61, *s6* 35–40, *r2* 60–62, *r3* 130–137, *r4* 17–19, *r5* 24–27. Seta *r6* smooth and off the shield. Opisthonotal shield with 22 to 23 pairs of thicked and pilose setae, of which setae *R* serie shortest.

*Ventral idiosoma* (Figure 8B). Tritosternum flanked by one pair of presternal platelets. Sternal setae *st1* (65–67) longer than *st2* (48–50) and *st3* (50–54). Endogynium (Figure 8C) with a saccate structure, its centre having a floriform structure. Gland pores *gv2* with three openings. Opisthogastric shield bearing ten pairs of setae. Seta *JV4*, *ZV4*, *SV2*, *SV3* distally pilose. Opisthogastric soft cuticle with five pairs of pilose setae, of which *JV5* is the longest. Peritrematal groove length: 304–314. Lengths of setae on opisthogastric shield: *JV1* 44–46, *JV2* 40–44, *JV3* 45–46, *JV4* 43–45, *ZV1* 13–14, *ZV2* 43–45, *ZV3* 46–49, *SV1* 14–15, *SV2* 26–28, *SV3* 34–38, *pa* 21–23, *po* 20–22.

*Gnathosoma* (Figure 8D–F). Gnathotectum emerging from denticulate base. Deutosternal groove with 11 denticulate rows; setae *h1* (61–63), *h2* (48–52), *h3* (49–54) smooth, and *pcx* (62–66) pilose. Fixed and movable digits of chelicera with several small teeth. Palp length: 271–281; trochanter bearing two stout setae (*v1* and *v2*); femur with five setae, of which *al* pectinate, *d3* sligthly pilose; genu with six pairs of setae, of which *al1* and *al2* entire.

*Legs*. Lengths of legs I–IV: 854–874, 600–622, 553–569, 823–855, respectively. Most leg setae distally pilose.

**Male** (n = 3). *Dorsal idiosoma* (Figure 9A). Idiosoma length: 683–703; width: 436–470. Holodorsal shield covering entire dorsum; a suture closely anterior to seta *J1*. All setae on the shield; setal form as in female.

*Ventral idiosoma* (Figure 9B). Tritosternum base reduced. Genital opening flanked by one pair of presternal platelets, sometimes fragmented. Lengths of setae *st1*–*st5* on sternogenital region: *st1* 59–61, *st2* 51–53, *st3* 42–44, *st4* 39–41, *st5* 37–40. Opisthogastric region with 12 pairs of setae, of which 7 pairs pilose. Peritrematal groove length 287–303.

*Gnathosoma* (Figure 9C–F). Movable digit with a big blunt tooth and several small teeth. Palp length: 212–239; trochanter with one pointed ventral protuberance, bearing seta *v2* evidently stouter, *v1* near the base. Lengths of setae: *h1* 25–29, *h2* 34–39, *h3* 48–50 and *pcx* 56–59.

*Legs.* Lengths of legs: I: 755–799; II: 514–553; III: 469–501; IV: 728–761. Femur II with a main spur (proximal) and an axillary process (distally); genu II and tibia II with a spur (Figure 9G).

**Other stages**. Unknown.

**Etymology.** The new species name is derived from the type locality Guizhou Province (*guizhouensis*). 

**Differential diagnosis.** *C. guizhouensis* sp. nov. is morphologically similar to *C. tibetensis* sp. nov. in the setal form with regard to the opisthonotal shield, the setal number on the opisthogastric shield of female and opisthogastric soft cuticle and the genu II and tibia II of male with a spur. However, *C. guizhouensis* sp. nov. is different from *C. tibetensis* sp. nov. as follows: (1) dorsal seta *z2* smooth and *s6* distally pilose, vs. seta *z2* distally pilose and *s6* smooth in *C. tibetensis* sp. nov.; (2) gnathotectum emerging from denticulate base, vs. nude base in *C. tibetensis* sp. nov.; (3) seta *pcx* on gnathosoma pilose, vs. smooth in *C. tibetensis* sp. nov. In addition, the differences between them in the female are as follows: (1) presternal platelets one pair, vs. three pairs in *C. tibetensis* sp. nov.; (2) opisthonotal shield with four pairs of pilose setae, vs. one pair in *C. tibetensis* sp. nov.; (3) seta *v1* and *v2* on palptrochanter stout and equal in length, vs. seta *v1* slender and about twice as long as seta *v2* in *C. tibetensis* sp. nov. The difference between them in the male is as follows: (1) opisthogastric region with eight pairs of pilose setae, vs. six pairs in *C. tibetensis* sp. nov.

#### 3.1.4. *Multidentatus* Jin & Yao sp. nov.

**Type material.** Holotype: ♀, encountered in rotten wood, Tianmushan National Nature Reserve (293 m a.s.l., N: 30°19′3″, E: 119°26′36″), Hangzhou, Zhejiang Province, China, 20 July 2018. Paratypes: two ♀♀ and one ♂, same collection data as the holotype; four ♀♀ and two ♂♂ encountered in decaying leaves, Tianmushan National Nature Reserve (754 m a.s.l., N: 30°20′10″, E: 119°27′5″), Hangzhou, Zhejiang Province, China, 20 July 2018; two ♀♀ and three ♂♂ encountered in rotten wood, Mangdangshan National Nature Reserve (619 m a.s.l., N: 26°20′42″, E: 119°26′13″), Nangping, Fujian Province, China, 6 August 2018.

**Diagnosis.** Both sexes: dorsal setae *z1*, *s1*, *s2*, *s6*, *r4*, *r5*, *R1*, *R2*, *R3* smooth; gnathotectum emerging from the nude base; seta *pcx* on gnathosoma smooth; setae *v1* on palptrochanter longer than *v2*. In the female, endogynium with a saccate structure, distal with an inverted V-shaped structure, the base with several teeth, each side flanked with two lamellar structures; opisthogastric shield bearing nine pairs of ventral setae, of which four pairs are distally pilose. In the male, opisthogastric region with eight pairs of distally pilose setae; seta *v2* on palptrochanter modified to button-shaped; movable digit with a big blunt; genu II and tibia II without spur.


**Description**


**Female** (n = 9). *Dorsal idiosoma* (Figure 10A). Idiosoma well sclerotized; length: 760–795: width: 559–576. Podonotal shield with 22 pairs of setae, of which setae *z1*, *s1*, *s2*, *s6*, *r4*, *r5* are fine and smooth, and others are thick and distally pilose. Lengths of setae on podonotal shield: *j1* 65–67, *j2* 75–77, *j3* 76–78, *j4* 114–118, *j5* 55–58, *j6* 75–77, *z1* 48–51, *z2* 65–68, *z3* 81–83, *z4* 83–87, *z5* 95–99, *z6* 65–67, *s1* 22–23, *s2* 20–24, *s3* 81–84, *s4* 89–92, *s5* 98–101, *s6* 19–23, *r2* 76–78, *r3* 160–165, *r4* 19–21, *r5* 22–24. Opisthonotal shield with 23 to 25 pairs of setae, of which *R1, R2* and *R3* are smooth, and others are distally pilose. Setae *R* series shortest.

*Ventral idiosoma* (Figure 10B). Tritosternum flanked by three pairs of presternal platelets. Sternal setae *st1* (62–75) and *st2* (60–62) stouter than *st3* (64–67). Gland pores *gv2* invisible. Endogynium with a saccate structure, distal with an inverted V-shaped structure, the base with several teeth, each side flanked with two lamellar structures (Figure 10C). Opisthogastric shield bearing nine pairs of ventral setae. Setae *JV4*, *ZV2*, *ZV3*, *SV2* distally pilose. Opisthogastric soft cuticle with three pairs of setae, of which *JV5* long (78–81) and distally pilose. Peritrematal groove length: 303–321. Lengths of setae on opisthogastric shield: *JV1* 61–63, *JV2* 62–65, *JV3* 65–69, *JV4* 65–68, *JV5* 85–89, *ZV1* 16–19, *ZV2* 65–67, *ZV3* 60–64, *SV1* 19–21, *SV2* 58–62, *pa* 27–29, *po* 22–25.

*Gnathosoma* (Figure 10D–F). Gnathotectum with three prongs, emerging from nude base. Fixed and movable digits of chelicera with several teeth. Deutosternal groove with ten denticulate rows; setae *h1* (69–74), *h2* (65–67), *h3* (84–92) and *pcx* (76–79) smooth. Palp length: 232–261; trochanter bearing two stout setae (*v1* and *v2*); femur with five setae (*al*, *d1*, *d2*, *d3* and *pl*), of which *al* pectinate and *d3* slightly pilose; genu with six pairs of setae *(al1*, *al2*, *d1*, *d2*, *d3 and pl*), of which *al1* and *al2* entire.

*Legs*. Lengths of legs I–IV: 787–860, 575–602, 617–647, 723–931, respectively. Most leg setae distally pilose.

**Male** (n = 3). *Dorsal idiosoma* (Figure 11A). Idiosoma length: 746–768; width: 532–545. Holodorsal shield covering entire dorsum; a suture closely anterior to seta *J1*. All setae on the shield; setal form as in female.

*Ventral idiosoma* (Figure 11B). Tritosternum base reduced. Genital opening flanked by three pairs of presternal platelets. Lengths of setae *st1*–*st5* on sternogenital region: *st1* 61–68, *st2* 64–67, *st3* 57–52, *st4* 61–64, *st5* 58–62. Opisthogastric region with 13 pairs of setae, of which 3 pairs distally pilose. Peritrematal groove length: 237–299.

*Gnathosoma* (Figure 11D–F). Gnathotectum with four prongs. Palp length: 239–245; trochanter with one pointed ventral protuberance, bearing seta *v2* button-shaped, and *v1* near the base. Movable digit with a blunt tooth. Lengths of setae: *h1* 63–67, *h2* 51–60, *h3* 72–74 and *pcx* 68–71.

*Legs.* Lengths of legs: I 755–799, II 514–553, III 469–501, IV 728–761. Femur II with a main spur (proximal) and an axillary process (distally); genu II and tibia II without spur (Figure 11G).

**Other stages.** Unknown.

**Etymology.** This species is named after its endogynium with many teeth (*multidentatus*). 

**Differential diagnosis.** *C. multidentatus* sp. nov. is morphologically similar to *C. tibetensis* sp. nov. in setal form and number on podonotal shield, the number of chelicera and presternal platelets, and to the setal form of subcapitulum. However, the differences between them in the female are as follows: (1) dorsal setae *R1*, *R2* and *R3* smooth, vs. pilose in *C. tibetensis* sp. nov.; (2) opisthogastric shield with nine pairs of setae, vs. ten pairs in *C. tibetensis* sp. nov.; (3) ventral seta *ZV2*, *ZV3* and *SV2* distally pilose, vs. smooth in *C. tibetensis* sp. nov.; (4) endogynium with several teeth, vs. without teeth in *C. tibetensis* sp. nov. The differences between them in the male are as follows: (1) opisthogastric region with three pairs of pilose setae, vs. six pairs in *C. tibetensis* sp. nov.; (2) gnathotectum with four prongs, vs. three prongs in *C. tibetensis* sp. nov.; (3) seta *v2* on palptrochanter button-shaped, vs. acicular in *C. tibetensis* sp. nov.; (4) genu II and tibia II without spur, vs. a spur in *C. tibetensis* sp. nov.

### 3.2. Key to the Species of the Genus Cerogamasus gen. nov.


**Females**


1. Opisthogastric shield with five pairs of setae; seta *v2* on the palptrochanter distally pilose setae………………………..…………………………..…….*C. coreanus* comb. nov.

– Opisthogastric shield with nine or ten pairs of setae; seta *v2* on the palptrochanter smooth……………………………………….……………………………………………..2

2. Opisthogastric shield with nine pairs of setae………………………………………....3

– Opisthogastric shield with ten pairs of setae…………………………………..……....4

3. Endogynium with several teeth; dorsal setae *R1*, *R2* and *R3* smooth................................................................................................*C. multidentatus* sp. nov.

– Endogynium without teeth; dorsal setae *R1*, *R2* and *R3* pilose. *C. anshunensis* sp. nov.

4. Dorsal seta *z2* pilose, *z2* smooth; opisthogastric shield with one pair of pilose setae………………………………………………………………….….*C. tibetensis* sp. nov.

– Dorsal seta *z2* smooth, *z2* pilose; opisthogastric shield with four pair of pilose setae…………………………………………………………….…*C. guizhouensis* sp. nov.


**Males**


1. Genu II with a spur; movable digit only with a big tooth……………............................2

– Genu II without spur; movable digit with a big tooth and several small teeth…........3

2. Tibia II with a spur; dorsal setae *R1*, *R2* and *R3* smooth…….. *C. multidentatus* sp. nov.

– Tibia II without spur; dorsal setae *R1*, *R2* and *R3* pilose……….*C. anshunensis* sp. nov.

3. Dorsal seta *z2* pilose, *z2* smooth; opisthogastric shield with six pairs of pilose setae………………………………………………………...............….*C. tibetensis* sp. nov.

– Dorsal seta *z2* smooth, *z2* pilose; opisthogastric shield with eight pairs of pilose setae……………………………………………………....………....*C. guizhouensis* sp. nov.

## 4. Discussion

The family Parasitidae comprises two subfamilies: Parasitinae and Pergamasinae. The generic concept of this family is not stable. The number of genera varies depending on authors and their view on the systematics of the family, especially on the rank of taxa [34]. The most comprehensive monograph on mites of Parasitidae contains 45 genera (23 genera within Parasitinae and 22 genera within Pergamasinae [7]). Juvara-Bals [8] described a new genus: *Occigamasus* Juvara-Bals, 2019 (Pergamasinae). Makarova [9] described another genus: *Thalassogamasus* Makarova, 2019 (Parasitinae). Yao et al. [10] indicated *Taiwanoparasitus* Tseng, 1995, a newly synonymized with *Psilogamasus*. The family now contains 46 genera, Parasitinae and Pergamasinae, each with 23 genera. These genera are easily distinguished from the new genus by the female with two shields; dorsal idiosoma in both sexes bears more than 40 pairs of setae, of which fewer than 7 pairs of podonotal setae are smooth; seta *z5* of dorsal hexagon similar to *j5* and *j6* in form (pilose or distally pilose); seta *al* of palpfemur pectinate, *al1* and *al2* of palpgenu entire; gnathotectum trispinate; female peritrematal shields posteriorly free; male palptrochanter with one pointed ventral protuberance.

The placement of the species *Cycetogamasus coreanus* needs some discussion. This species was previously known only from the adult females collected from litter and moist black humus in Korea [32]. The original description of *C. coreanus* is not very detailed or adequately illustrated. Recently, *C. coreanus* has been redescribed [33]. The genus *Cycetogamasus* was established by Athias-Henriot with *Cycetogamasus diviortus* Athias-Henriot, 1967, as its type species [32]. The main characteristics of the females of this genus are the presence of the cingulum, gland pores *gv2* with one or two openings and a movable digit with three teeth. *Cycetogamasus diviortus* is different from those species, especially the type species *C. diviortus*, in the following characteristics: absence of the cingulum, many teeth on the movable digit of the chelicera and gland pores *gv2* with three openings or not seen [32,33]. The common features of female *C. coreanus* with *Cerogamasus* gen. nov. are the following: female peritrematal shields free posteriorly; dorsal setae *z5*, *j5* and *j6* pilose or distally pilose; less than seven pairs of podonotal setae smooth; trochanter of palp with one pointed ventral protuberance in the male. We conclude that *C. coreanus* must be included in the new genus *Cerogamasus*.

*Cerogamasus coreanus* (Athias-Henriot) comb. nov.

*Cycetogamasus coreanus* Athias-Henriot, 1980: 290 [32]; Keum et al. 2019: 16 [33].

The geographical distribution of *Cerogamasus* gen. nov. is currently limited to Asia, i.e., China, Republic of Korea and North Korea [32,33]. Because only one deutonymph was collected from the moss, we are not confident about the true habitat of the species in the genus. The apparent absence of the juveniles from the decaying leaves, weed pile or rotten wood of the adult habitat might suggest another habitat for development of juveniles [32,33]. If immatures also inhabit decaying leaves, moss, weed piles or rotten wood, their absence may be explained by two reasons: the collector picked up the large individuals (adults) which were easily detectable with the naked eye, or almost all immatures had already reached adulthood before the moment of sampling [3,35,36]. The chelicerae of the species in the genus have several small teeth and look very robust, which is more suitable for crushing solid food rather than sucking fluids [1,37,38]. Catching live individuals and rearing them in the laboratory would uncover the feeding habits of this species. 

The presence of a swollen protuberance on the venter of the palptrochanter in the male may be an adaptation for optimal feeding [1,36]. Having most of the dorsal shield setae and leg setae relatively stout and pilose distally may give the mite an advantage to move freely among decaying leaves, moss, weed piles and rotten wood [39,40].

## Figures and Tables

**Figure 1 animals-14-02260-f001:**
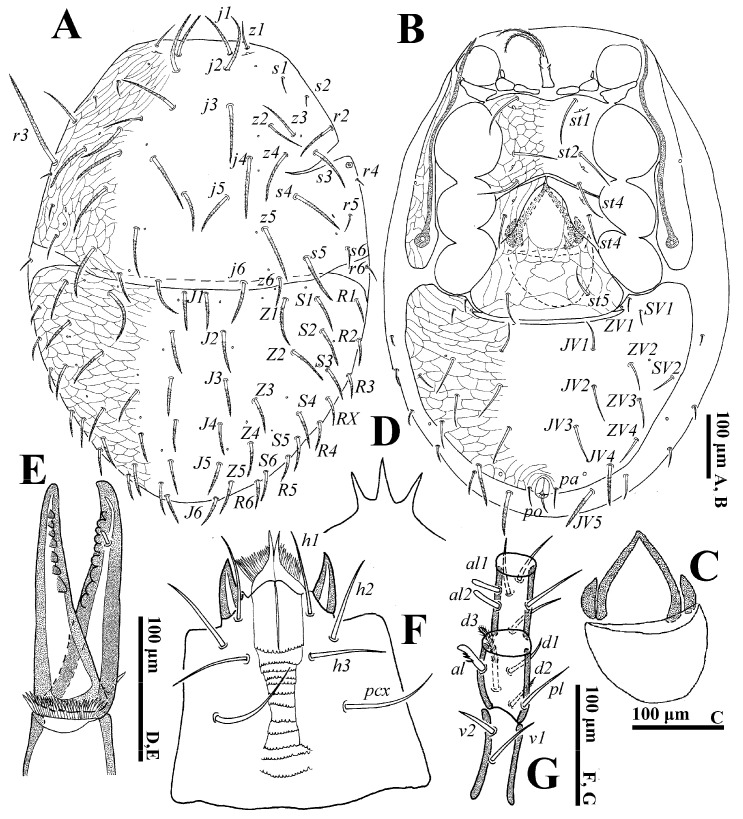
*Cerogamasus tibetensis* Jin & Yao sp. nov., female. (**A**)—Dorsal idiosoma. (**B**)—Ventral idiosoma. (**C**)—Endogynium. (**D**)—Gnathotectum. (**E**)—Chelicera. (**F**)—Subcapitulum. (**G**)—Trochanter, femur and genu of palp.

**Figure 2 animals-14-02260-f002:**
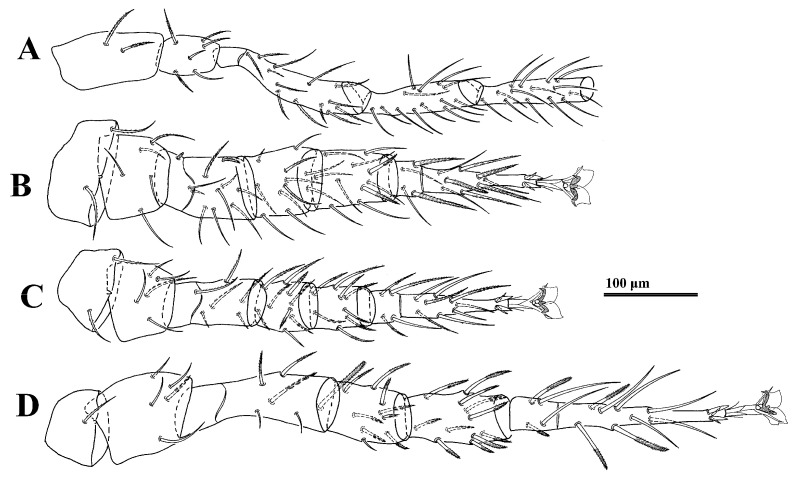
*Cerogamasus tibetensis* Jin & Yao sp. nov., female. (**A**)—Coxa–tibia of leg I. (**B**)—Leg II. (**C**)—Leg III. (**D**)—Leg IV.

**Figure 3 animals-14-02260-f003:**
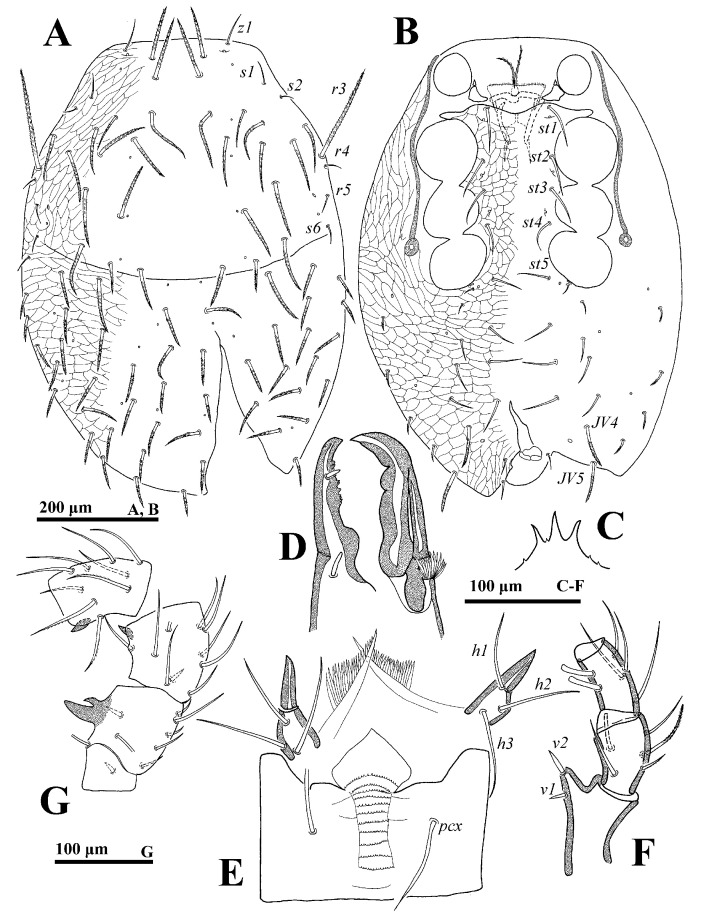
*Cerogamasus tibetensis* Jin & Yao sp. nov., male. (**A**)—Dorsal idiosoma. (**B**)—Ventral idiosoma. (**C**)—Gnathotectum. (**D**)—Chelicera. (**E**)—Subcapitulum. (**F**)—Trochanter, femur and genu of palp. (**G**)—Femur, genu and tibia of leg II.

**Figure 4 animals-14-02260-f004:**
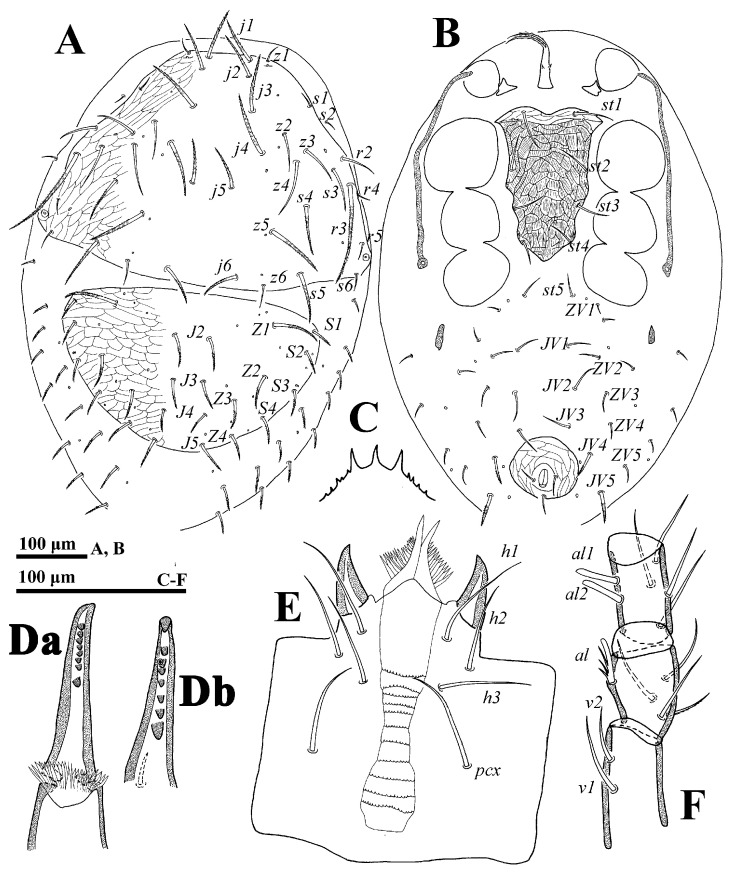
*Cerogamasus tibetensis* Jin & Yao sp. nov., deutonymph. (**A**)—Dorsal idiosoma. (**B**)—Ventral idiosoma. (**C**)—Gnathotectum. (**Da**)—movable digit of chelicera; (**Db**)—fixed digit of chelicera. (**E**)—Subcapitulum. (**F**)—Trochanter, femur and genu of palp.

**Figure 5 animals-14-02260-f005:**
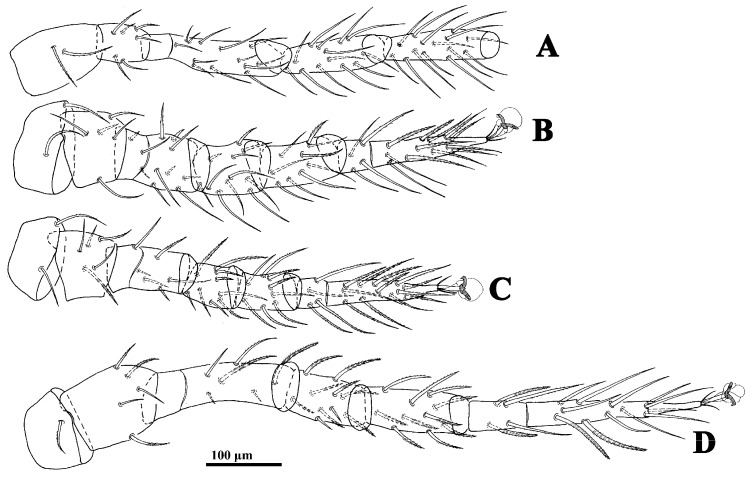
*Cerogamasus tibetensis* Jin & Yao sp. nov., deutonymph. (**A**)—Coxa–tibia of leg I. (**B**)—Leg II. (**C**)—Leg III. (**D**)—Leg IV.

**Figure 6 animals-14-02260-f006:**
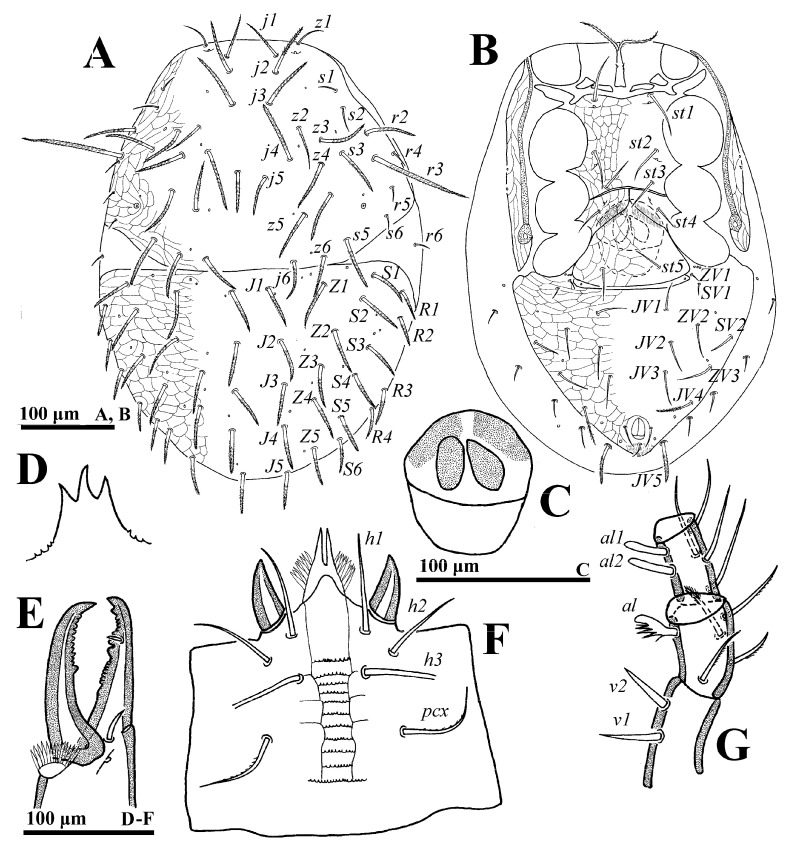
*Cerogamasus anhuiensis* Jin & Yao sp. nov., female. (**A**)—Dorsal idiosoma. (**B**)—Ventral idiosoma. (**C**)—Endogynium. (**D**)—Gnathotectum. (**E**)—Chelicera. (**F**)—Subcapitulum. (**G**)—Trochanter, femur and genu of palp.

**Figure 7 animals-14-02260-f007:**
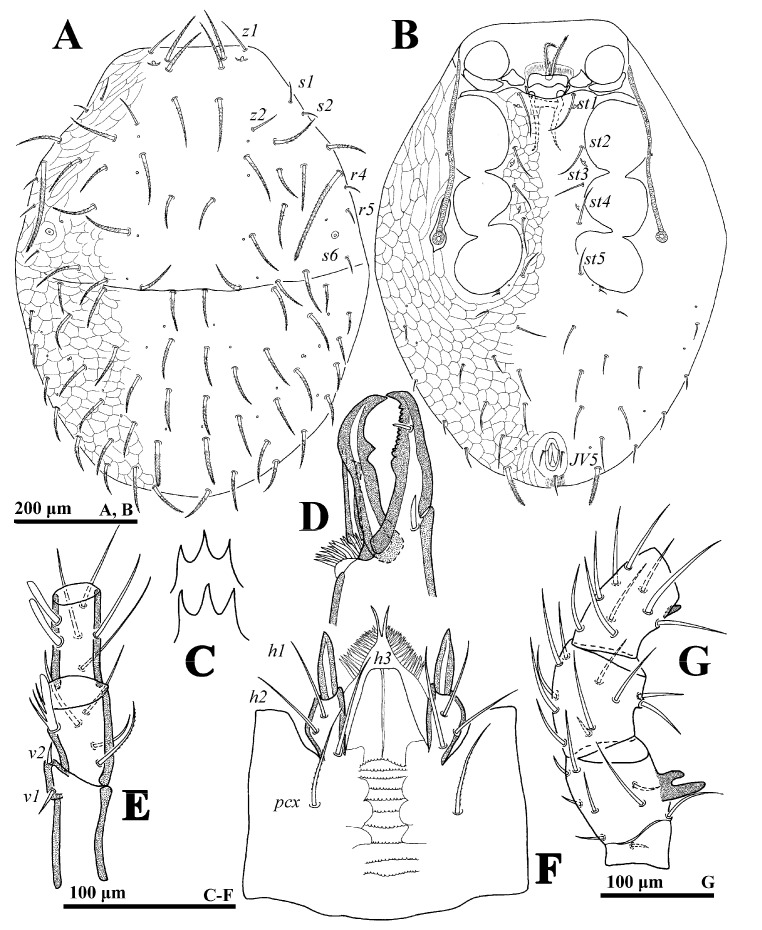
*Cerogamasus anhuiensis* Jin & Yao sp. nov., male. (**A**)—Dorsal idiosoma. (**B**)—Ventral idiosoma. (**C**)—Gnathotectum. (**D**)—Chelicera. (**E**)—Subcapitulum. (**F**)—Trochanter, femur and genu of palp. (**G**)—Femur, genu and tibia of leg II.

**Figure 8 animals-14-02260-f008:**
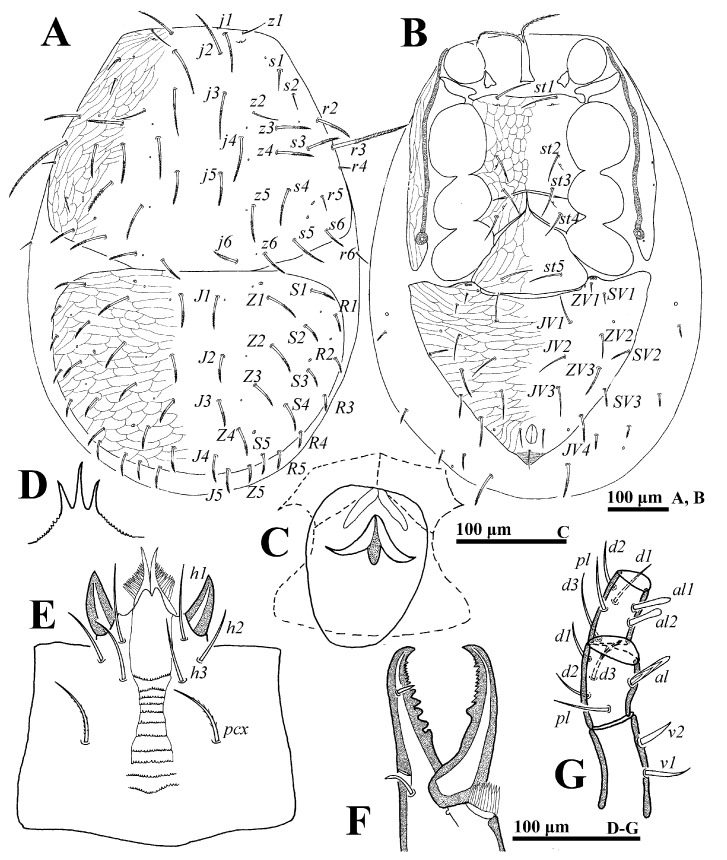
*Cerogamasus guizhouensis* Jin & Yao sp. nov., female. (**A**)—Dorsal idiosoma; (**B**)—Ventral idiosoma; (**C**)—Endogynium; (**D**)—Gnathotectum; (**E**)—Subcapitulum; (**F**)—Chelicera; (**G**)—Trochanter, femur and genu of palp.

**Figure 9 animals-14-02260-f009:**
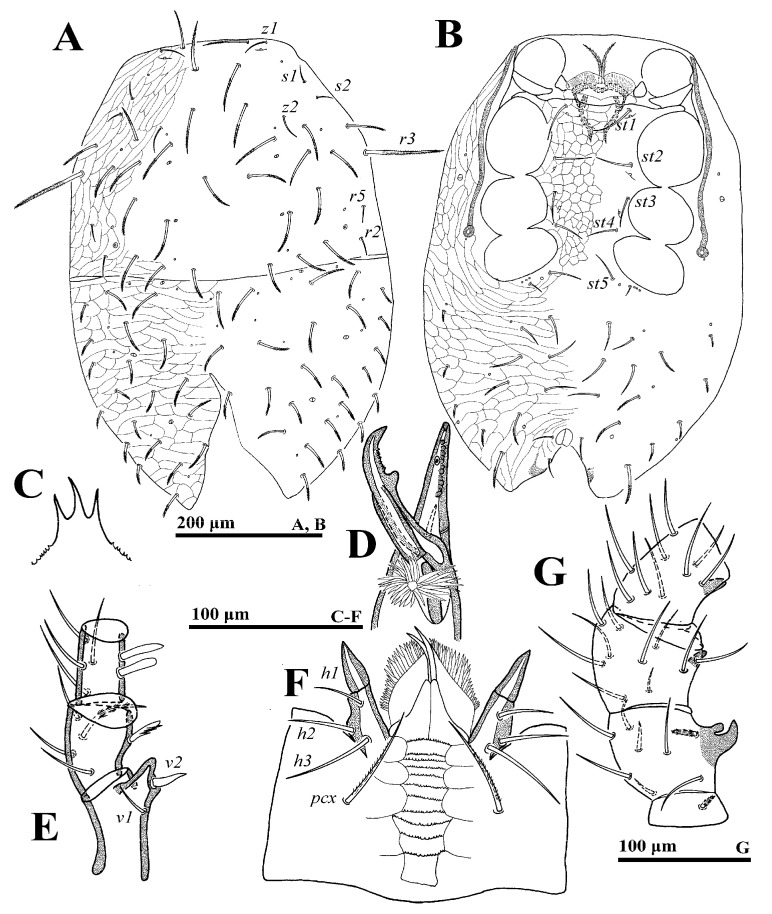
*Cerogamasus guizhouensis* Jin & Yao sp. nov., male. (**A**)—Dorsal idiosoma. (**B**)—Ventral idiosoma. (**C**)—Gnathotectum. (**D**)—Chelicera. (**E**)—Trochanter, femur and genu of palp. (**F**)—Subcapitulum. (**G**)—Femur, genu and tibia of leg II.

**Figure 10 animals-14-02260-f010:**
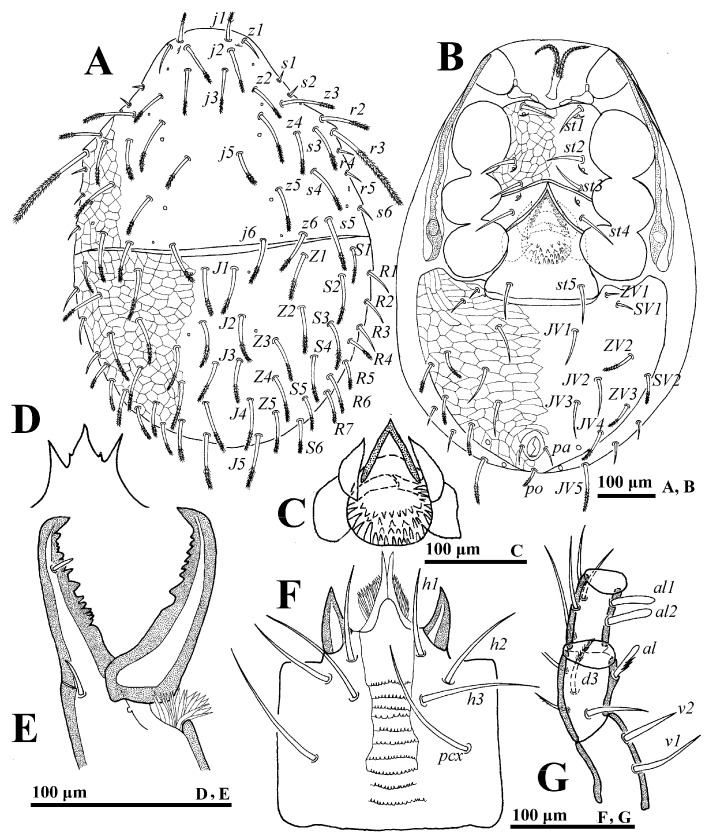
*Cerogamasus multidentatus* Jin & Yao sp. nov., female. (**A**)—Dorsal idiosoma. (**B**)—Ventral idiosoma. (**C**)—Endogynium. (**D**)—Gnathotectum. (**E**)—Chelicera. (**F**)—Subcapitulum. (**G**)—Trochanter, femur and genu of palp.

**Figure 11 animals-14-02260-f011:**
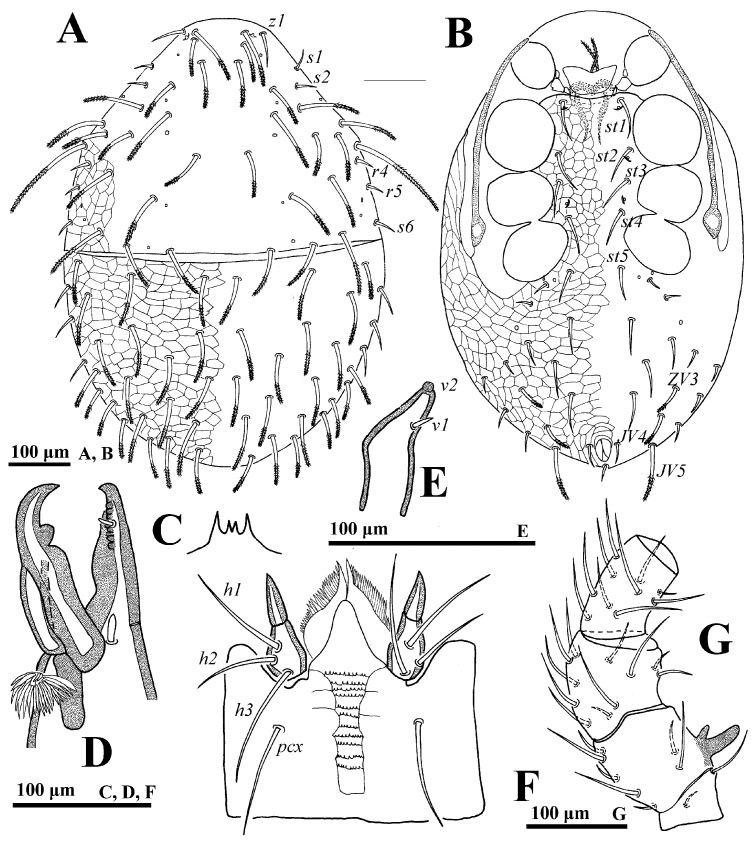
*Cerogamasus multidentatus* Jin & Yao sp. nov., male. (**A**)—Dorsal idiosoma. (**B**)—Ventral idiosoma. (**C**)—Gnathotectum. (**D**)—Chelicera. (**E**)—Trochanter of palp. (**F**)—Subcapitulum. (**G**)—Femur, genu and tibia of leg II.

## Data Availability

The data presented in this study are available on request from the corresponding author.

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
