# Peer review of "Cerogamasus, a New Genus of Parasitinae Mites, with Description of Four New Species from China (Acari: Parasitiformes: Parasitidae)"

_animals, 2024, doi:10.3390/ani14152260_

Round 1

Reviewer 1 Report

Comments and Suggestions for Authors

The authors described one new genus and four new species of Parasitidae mites, which contribute greatly to our knowledge of mite species diversity. All the descriptions and drawings are excellent. I here pointed out some minors.

(1) Line 22, "Protugamasus gen. nov. as new combination" should be revised as "Protugamasus gen. nov. as a new combination".

(2) Line 95, "A species, Cycetogamasus coreanus" revised as "Cycetogamasus coreanus".

(3) For the new combination, I suggest moving the description to the end following the description of the new species.

(4) Lines 116, 118, "Xizang Province" should be changed to "Tibet Autonomous Region".

Reviewer 2 Report

Comments and Suggestions for Authors

In the reviewed MS the authors describe a new genus of parasitine mites, give descriptions of four new species of this genus and state a new combination for a species transferred from another genus to this new genus. The MS contains only morphological data. Could the authors provide minimal DNA data on the taxa, e.g. Cox1 barcodes? This is easy do and it is important for further research in Mesostigmata. The MS in general is well written, however it needs moderate English polishing. Simple summary and Abstract need rewriting. The authors are requested to restructure the Introduction and Discussion. Finally, some important references for the methods and for the discussion parts of the MS are omitted. Additional remarks are below.

Simple summary: please rewrite this paragraph and make it really “simple”, currently it is written in scientific style and is hard to read for a unprepared reader

Abstract: some important data is absent, because it was given in Simple summary, which is suboptimal. Please, transfer distinct data from Simple summary to Abstract and rewrite the Simple summary.

25: Other a new species --- what do you mean saying this?

31: please add 1-2 more general sentences about Parasitiformes and Mesostigmata in order to prepare the sentences on Parasitidae.

31-34: could you please briefly explain why the family is called Parasitidae (referring to parasite), however, as you say, most members are predators (not parasites)

36-37: please explain briefly, what are the main similarity and dissimilarity between these genera, which led you to this conclusion? giving a list of traits may be the optimal way to do this

37: “superficially resemble Cycetogamasus Athias-Henriot, 1980 members, C. coreanus Athias-Henriot, 1980” --- not clear, what do you mean saying this?

40-42: redundant

47-49: this is a classical technique. Please, provide a reference for this method

47-48: please, say in general where and when the material was collected

49: Hoyer medium --- there are many different variations of this medium. Please, provide the distinct proportion of chemicals, which you used for preparing this medium

52-53: please specify which method of microscopy contrast did you use (Ph, DIC…)

57: adenotaxy and poroidotaxy --- please give in brackets a brief explanations of these terms

68: corniculi small --- please be more precise and explain better “small”: relative to which structure or give measurements

68: gnathotectum usually trispinate --- please, specify the shape if it is not trispinate

68: fixed digits of chelicera with more than seven teeth --- please specify ranges

81: it is probably better to rename this paragraph into “Differential diagnosis” and provide differences between the new genus and all other morphologically close parasitine genera. Please, say here directly which 2-3 parasitine genera are closest to the new species.

81: In the introduction you said that the new genus is close to Cycetogamasus. However here you compare it with another genus (Psilogamasus). It is confusing. Please provide comparison which several genera and say directly what the differences between them are. Please, reconsider section 3.1 following this remark on the line 81.

116: N 26°59′, E 100°11′ --- please provide more detailed gps coordinates. Revise this in the entire MS

Figure 1 (and other figures): please transfer the scale bar line closer to the corresponding figures as it is usually done in acarological papers.

Figure1 caption: please check if the format should be A -, B - , …  instead of A., B., …

Figure 3A,E – drawing a deformed specimen is suboptimal. Could you provide a drawing of undamaged mite?

Figures 1,2,3,4 – in all your drawings coxae 2.3.4 are fused and form a common figure consisting of 3 round segments. Is it correct? Please check in slides and confirm if they are fused or separated and specify this somewhere in the text.

Figure 5. Please add a blank line between the caption and the figure and check this for all your figures

200: for all new names (for all new species in the MS) please specify in the sections “Etymology”: part of speech, gender, origin

273: Differential taxonomy à Differential diagnosis (revise this in the entire MS)

296: Description - redundant

359: sp. nov.. --- remove the extra dot

444: gen. nov. --- bold

444: please provide a key for parasitine genera in order to place your new genus among them

468-488: some of the information given in these two paragraphs should be given in the Introduction because stylistically it does not belong to Discussion and the lack if this information in the beginning of the MS makes it harder to understand. Please, consider restructuring the Discussion and Introduction.

489-503: Did Dr Halliday participate in writing this concluding paragraph? Please provide references for different sentences in this paragraph.

Comments on the Quality of English Language

minor to moderate

Reviewer 3 Report

Comments and Suggestions for Authors

The manuscript is interesting. Four new species have been thoroughly described.

I would like to suggest some minor changes.

Some information regarding classification within the Parasitidae should be added to the Introduction. The authors mentioned only one subfamily Parasitinae, but the second subfamily is not mentioned in the text. 

The generic concept of the family Parasitidae is discussed and the authors should mention it in the text. 

There is also no information about the number of species that belong to the Parasitidae. Are there any endemic species among those found in China?

It is written in Discussion: "The taxonomy of Parasitidae is traditionally based on morphological characteristics of the female or deutonymph", but one of the key morphological characters in Parasitidae to separate genera is:   "palptrochanter of male with one pointed ventral protuberance". It is not clear to me. 

It is not clear to me what features distinguish the group of mentioned genera (Protugamasus gen. nov., Trachygamasus 486 Berlese, 1904, Psilogamasus Athias-Henriot, 1969 and Coprocarpais Hrúzová & Fenďa, 2018) from other genera of the family Parasitidae. In the previous sentence it is written that the mentioned features are: "key morphological characters in Parasitidae to separate genera". Are these features characteristic of all representatives of the Parasitidae or only for the specific group of those four genera?
